# Antimicrobial resistance profiles of bacteria from clinical specimens at Amhara Public Health Institute, Bahir Dar, Ethiopia: A retrospective study

Michael Getie[1]*, Wudu Tafere[1], Alem Tsega[1], Tsehaynesh Gebreyesus[1], Gizeaddis Belay[1], Alemayehu Abate[2], Hailu Getachew[1], Bayeh Abera[3], Demeke Endalamaw[1], Tazeb Molla[1], Teshiwal Deress[4], Belay Bezabih[5]

1 Medical Microbiology, Amhara Public Health Institute, Bahir Dar, Ethiopia, 2 Medical Biotechnology, Amhara Public Health Institute, Bahir Dar, Ethiopia, 3 Department of Medical Microbiology, College of Medicine and Health Sciences, Bahir Dar University, Bahir Dar, Ethiopia, 4 Department of Quality Assurance and Laboratory Management, School of Biomedical and Laboratory Sciences, College of Medicine and Health Sciences, University of Gondar, Ethiopia, 5 Field Epidemiology, Amhara Public Health Institute, Bahir Dar, Ethiopia

* mgetachew286@gmail.com

## Abstract

### Background

Antimicrobial resistance is a major public health problem worldwide, particularly in developing countries. The effectiveness of currently available antimicrobial is decreasing due to the increasing prevalence of resistant strains among bacterial isolates. This study aims to determine the antimicrobial resistance profile of bacterial isolates from different clinical specimens at Amhara Public Health Institute.

### Materials and methods

A retrospective analysis was conducted using data extracted from the WHONET 2024 database from July 1, 2022, to December 31, 2024, at the Amhara Public Health Institute bacteriology and mycology reference laboratory. The age range of the patients included in this study was 1–96 years, and the mean age of the patients was $33.44 \pm 17.36$ years. The data included demographic characteristics of patients, types of bacterial isolates and antimicrobial resistance profiles, which were analyzed using SPSS version 20 statistical software. The descriptive statistics were displayed as percentages and frequencies. The chi-square test was used to determine the prevalence of bacterial isolates from patients by age and sex. P-values below 0.05 were seen as a sign of a statistically significant difference.

**Data availability statement:** All relevant data are within the paper and its Supporting Information files.

**Funding:** The author(s) received no specific funding for this work.

**Competing interests:** The authors have declared that no competing interests exist.

**Abbreviations:** AMR, Antimicrobial Resistance; APHI, Amhara Public Health Institute; ATCC, American type culture collection; CLSI, Clinical and laboratory standard Institute; CoNS, Coagulase-negative staphylococcus; CSF, Cerebrospinal fluid; ESBL, Extended spectrum ß-lactamase; IRB, Institutional review board; MDR, Multi drug resistance; SOP, Standard operating procedure; SPSS, Statistical package for social sciences; WHO, World Health Organization

## Results

A total of 1165 specimens were processed, resulting in a culture-positive rate of 41% (478/1165) for bacterial pathogens isolated from clinical specimens. The majority of bacterial isolates were from stool (55%; 263/478), urine (20%; 96/478), wound (12.9%; 62/478), and blood (9.8%; 45/478), respectively. Of these, Gram-negative bacteria accounted for 89.1% (426/478) and Gram-positive bacteria accounted for 10.8% (52/478). The predominant bacterial isolates were *Vibrio cholerae* 54.6% (261/478), *E. coli* 16.1% (77/478), *Klebsiella* spp 6% (29/478), *S. aureus* 4.6% (22/478) and *Enterococcus* spp 2.9% (14/478). In this study *Proteus* spp 67.6% (46/68), *A. baumannii* 58.4% (31/53), and *Klebsiella* spp 64.1% (136/212) were identified as the most resistant bacteria to the tested antimicrobial. *S. aureus* shows resistance to tobramycin 100% (1) and penicillin 100% (17), oxacillin 84.6% (11/13) and tetracycline 63.6% (7/11). *Enterococcus* spp resistance to vancomycin 85.7% (6/7), penicillin 72.7% (8/11) and ampicillin 62.5% (5/8). In total, 53.1% (254/478) of the bacterial isolates were classified as multidrug-resistant (MDR), with 93.7% (238/ 254) being Gram-negative bacteria.

## Conclusions

Both Gram-negative and Gram-positive isolates showed high levels of resistance to commonly used antimicrobial. To address the problem of antimicrobial resistance, healthcare providers should focus on responsible antimicrobial prescribing practices based on local antibiogram data.

## Introduction

Antimicrobial resistance (AMR) is a phenomenon in which microorganisms become resistant to antimicrobial agents to which they were originally sensitive [1]. Antimicrobial resistance is a major Public health problem worldwide, particularly in developing countries where infectious diseases, poverty and malnutrition are endemic [2]. Recent data shows that about 700,000 death per year are attributable to AMR infections and projected to increase to 10 million annually by 2050 if the present trends persist [3].

Multidrug-resistant (MDR) bacteria are difficult to treat, limit therapeutic options, prolong hospitalization, require higher doses, and have higher tendencies for toxicity [4]. The slow progress in research and development of novel antimicrobial, due to the emergence of MDR pathogens [5].

The problem of antimicrobial resistance is not only the cause of the development of resistance but also the transmission of the resistant strains from one person to another, especially in a health facility setting [6]. The problem worsens in Ethiopia, due to multiple factors including lack of surveillance systems, limited resources, poor infection prevention and control practice, misuse and overuse of antimicrobial, and lack of clinical microbiology laboratories to identify the specific etiologic agents and their antimicrobial susceptibility testing has increased empirical therapy, which in turn leads to the emergence of AMR [7].

In Ethiopia, identifying the most common bacterial pathogens and their resistance patterns is crucial to optimizing therapy and ultimately reducing the morbidity and mortality linked to infectious diseases [8]. This study aimed to identify the bacterial pathogens and their antimicrobial resistance profiles in clinical specimens sent to the Amhara Public Health Institute (APHI).

## Materials and methods

### Study area

The study was conducted at Amhara Public Health Institute, Bahir Dar city, Amhara Regional State, Ethiopia which is approximately 565 km away from the capital city, Addis Ababa. The town has a latitude of 12°361 N and a longitude of 37°281 E with an elevation of 2133 meters above sea level. The institute provides healthcare services to over twenty-five million people in the region. It has an accredited reference level laboratory with 7 sections and a separate reception room. It was accredited by the Ethiopian National Accreditation Office. The microbiology section is one of the principal areas; it is estimated that 1,600 clinical specimens are delivered annually. This section provides accredited diagnostic laboratory testing services, including microscopy, culture, organism identification, and antimicrobial susceptibility testing (AST) for patients or specimens referred to from zonal and regional health facilities.

### Study design and period

An institution-based retrospective study was conducted by accessing the APHI bacteriology and mycology reference laboratory WHONET 2024 database from July 1, 2022, to December 31, 2024.

### Study population

The dataset comprised 1,165 patient specimens with suspected bacterial infections. Demographic characteristics patients and culture results were accessible in the APHI bacteriology and mycology reference laboratory WHONET 2024 database from July 1, 2022, to December 31, 2024.

### Sampling methods

The study used a comprehensive sampling method that incorporated all bacteriological culture records of patients of any age who were suspected of having a bacterial infection during the study period.

### Inclusion and exclusion criteria

All specimen entries on the WHONET 2024 database during the study period, having information on the age of a patient, sex, source of specimen, type of specimen, hospital units, isolated organism, and antimicrobial resistance profile, were included in this study. However, entries without any of the aforementioned information or specimens with unknown specimen type, unknown source of specimen, and specimens without culture result status were excluded from this study.

### Bacterial isolation and identification

The standard operating procedures (SOPs) of specimen collection and transportation of different clinical specimens were implemented. The collected clinical specimens were delivered to the bacteriology and mycology reference laboratory and processed following standard procedures. Conventional microbiological culture methods were employed to isolate and identify bacteria. Media was prepared in-house as per procedures stipulated in Cheesbrough [8]. Clinical specimens, including urine, blood, sputum, wound/pus, cerebrospinal fluid, body fluid, discharge (ear/eye), throat, and sputum, were cultured. Each clinical sample employed standard microbiological culturing techniques. Specimens were inoculated into the appropriate isolation culture media and incubated at 35–37 °C, according to standard protocols for each sample.

Bacterial identification was made mainly based on colony characteristics, Gram stain reaction, and proper biochemical tests as per suitability according to CLSI guidelines [9] and developed SOPs. Identification of Gram-positive bacteria was done using Gram stain, hemolytic activity on sheep blood agar plates, catalase reaction, and coagulase test. Gram-negative bacteria were identified based on colony morphology on blood agar and MacConkey agar, followed by biochemical reactions, namely oxidase, triple sugar iron (TSI), sulphur indole and motility (SIM), citrate, lysine decarboxylase (LDC), and urease tests. After bacterial identification, antimicrobial susceptibility tests were done on Mueller-Hinton agar (Oxoid Basingstoke, UK) using the Kirby-Bauer disk diffusion method [10].

## Antimicrobial susceptibility testing

Antimicrobial susceptibility testing of the isolates was performed by the Kirby–Bauer disk diffusion test method on Mueller–Hinton agar for the following antimicrobial agents (Oxoid, Basingstoke, Hampshire, UK) [11,12]. Standard antimicrobial discs with specified concentrations were used to detect the resistance patterns of each isolate. The plates were incubated overnight. After incubation was completed, the zone inhibition diameter was measured in millimeters. The zones were interpreted as susceptible, intermediate, or resistant according to CLSI 2024 [9]. However, during antimicrobial susceptibility testing and reporting considerations for each organism group include agents of proven test efficacy that show acceptable in vitro test susceptibility and effective clinically be analyzed and reported as susceptible. The definition of CDC was used in this study for MDR: resistance of bacterial isolates to at least one antimicrobial in three or more drug classes [13]. The following standard antimicrobial, with abbreviated names and disk contents in brackets, were used to test the resistance profiles of bacterial isolates: Gram-positive isolates were tested for ampicillin (AMP) (10 μg), cefoxitin (FOX) (30 μg), clindamycin (DA) (2 μg), ciprofloxacin (CIP 5 μg), ceftriaxone (CRO) (30 μg), chloramphenicol (CHL) (30 μg), erythromycin (ERY) (15 μg), gentamicin (GEN) (10 μg), penicillin (PEN) (10 μg), nitrofurantoin (NIT) (300 μg), trimethoprim-sulphametazol (SXT) (1.25/23.75 μg), tetracycline (TCY) (30 μg), tobramycin (TOB) (10 μg) and vancomycin (Van) (30 μg) [9,14]. Gram-negative isolates were tested for ampicillin (AMP) (10 μg), amoxicillin-clavulanic acid (AMC) (20/10 μg), ceftazidime (CAZ) (30 μg), ceftriaxone (CRO) (30 μg), ciprofloxacin (CIP) (5 μg), chloramphenicol (CHL) (30 μg), gentamicin (GEN) (10 μg), meropenem (MEM) (10 μg), tobramycin (TOB) (10 μg), trimethoprim-sulphmetaxzole (SXT) (1.25/23.75 μg)and tetracycline (TCY) (15 μg) [9,14].

## Data source and access

Data for this study were obtained from the APHI bacteriology and mycology reference laboratory, WHONET 2024, a software database tool developed by the World Health Organization for antimicrobial resistance surveillance [15]. This electronic database included all clinical culture data collected from July 1, 2022, to December 31, 2024. The data was accessed on January 6, 2025, exclusively for this study. The data accessed was aggregated through Microsoft Excel 2013 and analyzed using SPSS version 20.

## Data quality control

Standard operating procedures for bacteriological techniques were followed throughout clinical specimen collection, transportation, culture media preparation, bacterial isolation, identification and antimicrobial susceptibility testing. Culture media sterility was ensured by random selection and incubation of 5% of prepared media. Media performance was regularly evaluated using known standard strains of *E. coli* (ATCC25922), *S. aureus* (ATCC25923) and *P. aeruginosa* (ATCC 27853). The WHONET 2024 database offers a number of quality control systems and alerts to check data quality.

## Data analysis

Data extracted from the WHONET 2024 database [16], aggregated through a Microsoft Excel 2013 spreadsheet for cleaning and validation, and then transferred to SPSS version 20 software for analysis. Descriptive statistics were used to

designate the demographic characteristics of the participants, magnitudes of bacterial isolates and antimicrobial resistance profiles of the isolates. Chi-square test was employed to reveal age and sex specific prevalence of bacterial isolates from patients. P-value of less than 0.05 was considered to indicate statistically significant difference. Finally, the results of the findings were presented as frequency and percentage in texts, tables, and graphical forms.

### Ethics approval

This manuscript does not involve the use of data from any animal or bio specimens from deceased individuals. This retrospective study using WHONET 2024 database, the IRB fully waived the requirement for informed consent. We obtain permission from the relevant laboratory diagnostic directorate and APHI general director. The regional public health research ethical review committee granted ethical approval in May 2024 under the number NoH/R/T/T/D/07/74.

## Results

### Demographic characteristics of patients with bacterial isolates

A total of 1165 specimens were included in this study that met the eligibility criteria from different clinical specimens (stool, urine, blood, wound, genital/ urethral, discharge (ear/eye), and CSF. The mean age of the patients was 33.44 ± 17.36 years, and male patients accounted for 54.1% (631/1165). The age of the study participants ranged from 1 day to 96 years. Most of the study participants were between the ages of 15 and 64 years, 78.2% (912/1165). Most specimens 71.2% (829/1165) were from the medical outpatient department (MOPD), 25.7% (299/1165) from the emergency outpatient department (EOPD), 20.9% (244/1165) from the inpatient ward, and 1.1% (13/1165) specimens were from the community (Table 1).

Overall, 41% (478/1165) of the specimens were positive for aerobic bacterial isolates. Of these, 89.1% (426/478) were Gram-negative bacteria and 10.8% (52/478) were Gram-positive bacteria and the highest isolation rates were obtained from 15–64 years age group 77.2% (369/478) not statistically significant (P = 0.424).The isolated bacteria was relatively higher in males 56.1% (268/478) than females 43.9% (210/478) though not statistically significant (p = 0.26).The most frequently culture-processed specimens were stool 41.7% (486/1165) and urine 31.7% (369/1165) (Table 1 and Table 2).

### The magnitudes of bacterial isolates

The most frequently isolated Gram-negative bacteria were *Vibrio cholerae* 54.6% (261/478) and *E. coli* 16.1% (77/478). *S. aureus* 4.6% (22/478) and *Enterococcus* spp 2.9% (14/478) were the predominant isolated Gram-positive bacteria. Most of the bacterial isolates were from stool specimens, 55.4% (265/478) followed by urine, 20.5% (98/478) and wound, 12.1% (58/478) (Table 3).

### Antimicrobial resistance in Gram-negative bacteria

In this study, *Proteus* spp 67.6% (46/68), *A. baumannii* 58.4% (31/53), and *Klebsiella* spp 64.1% (136/212) were identified as the most resistant bacteria to the commonly used antimicrobial. These bacteria exhibited resistance to ceftazidime, ciprofloxacin, and trimethoprim-sulfamethoxazole. Specifically, *Proteus* spp showed 85.7% (6/7) resistance to ceftazidime, 75% (6/8) to ciprofloxacin, and 75% (6/8) to trimethoprim-sulfamethoxazole. *A. baumannii* resistance rate of 77.8% (7/9) to ceftazidime, 62.5% (5/8) ciprofloxacin and 40% (2/5) to trimethoprim-sulfamethoxazole. *Klebsiella* spp displayed 64% (16/25) resistance to ceftazidime, 61.5% (5/8) to ciprofloxacin, and 76.2% (16/21) to trimethoprim-sulfamethoxazole. Furthermore, *E. coli*, *Proteus* spp, and *Vibrio cholerae* were resistant to ampicillin, 87.7% (57/65), 87.5% (6/7) and 99.2% (122/123), respectively (Fig 1).

Antimicrobial resistance (AMR) profile of Gram-negative bacteria showed that erythromycin 99.1% (116/117), ampicillin 94% (188/200), trimethoprim-sulfamethoxazole 82.1% (188/229), amoxicillin-clavulanic acid 81.5% (66/81), ceftazidime 54.7% (80/146) and tetracycline 51.5% (17/33) was substantially resisted antimicrobial (Table 4).

**Table 1. Patient profiles, clinical specimens, hospital unites, and culture results.**

| Variables | | Frequency | Percent |
|---|---|---|---|
| Sex | Male | 631 | 54.2 |
| | Female | 534 | 45.8 |
| Age groups | 0-4 | 104 | 8.9 |
| | 5 −14 | 78 | 6.7 |
| | 15-64 | 912 | 78.2 |
| | 65+ | 71 | 6.1 |
| Hospital units | Medical OPD | 829 | 71.2 |
| | Emergency OPD | 299 | 25.7 |
| | Inpatient Ward | 24 | 2.1 |
| | Other | 13 | 1.1 |
| Clinical specimens | Stool | 486 | 41.7 |
| | Urine | 369 | 31.7 |
| | Blood | 155 | 13.3 |
| | Wound/pus | 107 | 9.2 |
| | Genital/Ureteral | 16 | 1.4 |
| | Discharge (ear/eye) | 16 | 1.4 |
| | Cerebrospinal fluid | 9 | 0.8 |
| | Body fluid | 5 | 0.4 |
| | Sputum | 1 | 0.9 |
| | Throat | 1 | 0.9 |
| Gram stain results | Gram-negative bacteria | 426 | 88.4 |
| | Gram-positive bacteria | 52 | 10.8 |
| | Candida albicans | 4 | 0.8 |
| | No growth | 683 | 58.6 |

Other-community, OPD-out patient department.

**Table 2. Age and sex specific prevalence of bacterial isolates from patients.**

| Variables | Frequency | Culture positives | Prevalence (%) | $x2$ (p. value) |
|---|---|---|---|---|
| Sex | | | | 1.183 (0.26) |
| Male | 631 | 268 | 56.1 | |
| Female | 534 | 210 | 43.9 | |
| **Age groups** | | | | 2.971 (0.424) |
| 0-4 | 104 | 48 | 10 | |
| 5-14 | 78 | 28 | 5.9 | |
| 15-64 | 912 | 369 | 77.2 | |
| 65+ | 71 | 33 | 6.9 | |

$x2$- chi-square.

## Antimicrobial resistance profiles of Gram-positive bacteria

Gram-positive bacteria showed resistance to tobramycin 100% (1), vancomycin 91.6% (11/12), oxacillin 87.5% (14/16), penicillin 83.3% (25/30), ceftriaxone 66.6% (2/3) and tetracycline 60% (9/15). But remained highly susceptible to chloramphenicol 100% (5), nitrofurantoin 100% (22) and clindamycin 93.7% (15/16). Among Gram-positive bacteria, *S. aureus*

Table 3. The magnitudes of bacterial isolates from clinical specimen.

| Bacterial isolates | Clinical specimens | | | | | | | | Total |
|---|---|---|---|---|---|---|---|---|---|
| | Blood | | Discharge (ear/eye) | Genital | Cerebrospinal fluid | Stool | Urine | Wound | Wound |
| *A. baumannii* | 6 (1.2) | 0 | 0 | 0 | 0 | 1(0.2) | 3(0.6) | | 10 (2) |
| *Citrobacter* spp | 0 | 0 | 0 | 0 | 0 | 4 (0.8) | 2 (0.4) | | 6 (1.2) |
| *Klebsiella* spp | 5 (1) | 0 | 0 | 2 (0.4) | 1 (0.2) | 13 (2.7) | 8(1.6) | | 29 (6) |
| *Enterobacter cloacae* | 0 | 1 (0.2) | 0 | 0 | 0 | 1 (0.2) | 1 (0.2) | | 3 (0.6) |
| *E. coli* | 3 (0.6) | 1 (0.2) | 0 | 1 (0.2) | 1 (0.2) | 52 (10.8) | 19(3.9) | | 77(16.1) |
| *Enterococcus* spp | 11 (2.3) | 0 | 0 | 0 | 0 | 3 (0.6) | 0 | | 14 (2.9) |
| *Pseudomonas aeruginosa* | 0 | 1 (0.2) | 1 (0.2) | 2 (0.4) | 0 | 14 (2.9) | 7(1.4) | | 25 (5.2) |
| *Proteus* spp | 0 | 0 | 0 | 0 | 0 | 2 (0.4) | 8 (1.6) | | 10 (2) |
| *Salmonella typhi* | 0 | 0 | 0 | 0 | 2 (0.4) | 0 | 0 | | 2 (0.4) |
| *S. aureus* | 5 (1) | 2 (0.4) | 0 | 0 | 0 | 6 (1.2) | 9 (1.8) | | 22 (4.6) |
| *Vibrio cholera* | 0 | 0 | 0 | 0 | 261 (54.6) | 0 | 0 | | 261 .(54.6) |
| *Streptococcus viridians* | 5 (1) | 0 | 0 | 0 | 0 | 0 | 0 | | 5 (1) |
| *Others* | 12 (2.5) | 0 | 1 (0.2) | 0 | 0 | 0 | 1 (0.2) | | 14 (2.9) |
| Total | 47 (9.8) | 5 (1) | 2 (0.4) | 5 (1) | 265 (55.4) | 96 (20) | 58 (12.1) | | 478 (100) |

Key: Others include Coagulase-negative staphylococci, *Providencia* spp, *S.pyogenes*, *Haemophilus* spp and *Serratia* spp.

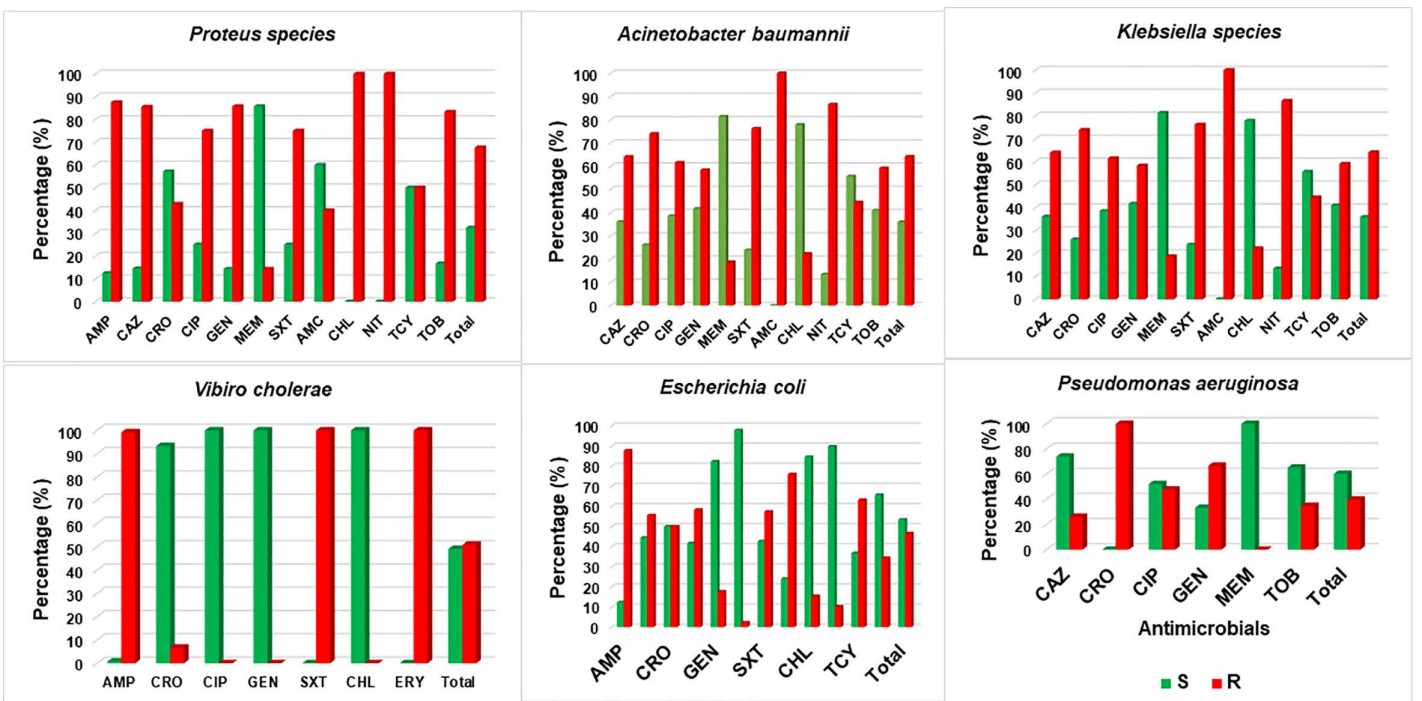

Fig 1. Antimicrobial resistance levels of the six major top isolated Gram-negative bacteria.

resistance to tobramycin 100% (1), penicillin 100% (17), oxacillin 84.6% (11/13), and tetracycline 63.6% (7/11). *Entero-coccus* spp exhibited resistance to vancomycin 85.7% (6/7), penicillin 72.7% (8/11) and ampicillin 62.5% (5/8). *S. viridians* were resistant to ceftriaxone 100% (2) and vancomycin 100% (5) seen in (Fig 2 and Table 5).

**Table 4. Antimicrobial resistance profile of Gram-negative bacterial isolates.**

| Bacteria | | AMP | CAZ | CRO | CIP | GEN | MEM | SXT | AMC | CHL | NIT | TCY | TOB | ERY | Total |
|---|---|---|---|---|---|---|---|---|---|---|---|---|---|---|---|
| A.baumannii | #T | NA | 9 | 5 | 8 | 10 | 8 | 5 | NA | NA | NA | NT | 8 | NA | 53 |
| | R (%) | NA | 7 (77.8) | 1 (20) | 5 (62.5) | 7 (70.0) | 3 (37.5) | 2 (40) | NA | NA | NA | NT | 6 (75) | NA | 31 (58.4) |
| Citrobacter spp | #T | NA | 4 | 3 | 4 | 4 | 4 | 3 | NT | 2 | 2 | NT | 4 | NA | 30 |
| | R (%) | NA | 4 (100) | - | 3 (75) | 2 (50) | - | 2 (66.7) | NT | 1 (50) | 2(100) | NT | 2 (50) | NA | 16 (53.3) |
| Klebsiella spp | #T | NA | 25 | 23 | 26 | 24 | 16 | 21 | 22 | 9 | 15 | 9 | 22 | NA | 212 |
| | R (%) | NA | 16 (64) | 17(73.9) | 16 (61.5) | 14 (58.3) | 3 (18.7) | 16 (76.2) | 22 (100) | 2 (22.2) | 13 (86.6) | 4 (44.4) | 13 (59.1) | NA | 136 (64.1) |
| Enterobacter cloacae | #T | NA | 3 | 3 | 3 | 3 | 2 | 2 | 2 | 1 | 1 | 2 | 2 | NA | 25 |
| | R (%) | NA | 1 (33.3) | 2 (66.6) | 1 (33.3) | 1 (33.3) | 1 (50) | 1 (50) | 2 (100) | 1(100) | 1 (100) | - | 2 (100) | NA | 13 (52) |
| E. coli | #T | 65 | 72 | 64 | 72 | 62 | 45 | 61 | 50 | 13 | 49 | 19 | 67 | NA | 639 |
| | R (%) | 57 (87.7) | 40 (55.6) | 32 (50) | 42 (58.3) | 11(17.7) | 1 (2.2) | 35 (57.4) | 38 (76) | 2 (15.4) | 5 (10.2) | 12(63.2) | 23 (34.3) | NA | 298 (46.6) |
| P.aeruginosa | #T | NA | 23 | 2 | 23 | 6 | 1 | NA | NA | NA | NA | NA | 23 | NA | 78 |
| | R (%) | NA | 6 (26.1) | 2 (100) | 11(47.8) | 4 (66.7) | - | NA | NA | NA | NA | NA | 8 (34.8) | NA | 31 (39.7) |
| Proteus spp | #T | 8 | 7 | 7 | 8 | 7 | 7 | 8 | 5 | 2 | 1 | 2 | 6 | NA | 68 |
| | R (%) | 7 (87.5) | 6 (85.7) | 3 (42.9) | 6 (75.0) | 6 (85.7) | 1 (14.3) | 6 (75.0) | 2 (40) | 2(100) | 1(100) | 1 (50) | 5 (83.3) | NA | 46 (67.6) |
| Salmonella typhi | #T | 2 | NT | NT | 2 | NA | NT | 2 | NT | NT | NA | NT | NA | NA | 6 |
| | R (%) | 2 (100.0) | NT | NT | 0 (0.0) | NA | NT | 1 (50.0) | NT | NT | NA | NT | NA | NA | 3 (50.0) |
| Vibrio cholerae | #T | 123 | NT | 119 | 120 | 1 | NT | 123 | NT | 122 | NT | NT | NT | 116 | 724 |
| | R (%) | 122(99.2) | NT | 8 (6.7) | 0 (0.0) | 0 (0.0) | NT | 123 (100) | NT | - | NT | NT | NT | 116 (100) | 369 (50.9) |
| Others | #T | 2 | 3 | 1 | 3 | 3 | 1 | 3 | 2 | 2 | NT | 1 | 2 | 1 | 28 |
| | R (%) | - | - | - | 2 (66.6) | 1 (33.3) | - | 2(66.6) | 2(100) | - | NT | - | - | - | 7 (25) |
| Total | #T | 200 | 146 | 228 | 272 | 120 | 84 | 229 | 81 | 149 | 68 | 33 | 134 | 117 | 1863 |
| | R (%) | 188 (94) | 80 (54.7) | 65 (28.5) | 86 (31.6) | 46 (38.3) | 9 (10.7) | 188 (82.1) | 66 (81.5) | 8 (5.4) | 22 (32.3) | 17 (51.5) | 59 (44.0) | 116 (99.1) | 950 (50.9) |

AMP-ampicillin, CAZ-ceftazidime, CRO-ceftriaxone, CIP-ciprofloxacin, GEN-gentamicin, MEM-meropenem, SXT- trimethoprim-sulfamethoxazole, AMC-amoxicillin-clavulanic acid, CHL-chloramphenicol, NIT-nitrofurantoin, TCY- tetracycline, TOB-tobramycin, ERY-erythromycin, NA-not analyzed (during antimicrobial susceptibility testing and reporting antimicrobial were selected based on CLSI, 2024 for each organism group but agents of proven test efficacy that show unacceptable in vitro test susceptibility and ineffective clinically should not be analyzed), NT- not tested (antimicrobial were not tested due unavailability of antimicrobial agents in bacteriology and mycology reference laboratory).

## Multidrug resistance profiles of bacterial isolates

In a total of 478 bacterial isolates, 53.1% (254) were identified as multidrug resistant (MDR). A significant proportion of these, 93.7% (238/254), were Gram-negative bacteria, while 6.3% (16/254) were Gram-positive bacteria. Among the MDR Gram-negative bacteria, *Proteus* spp, *A. baumannii*, *Klebsiella* spp and *E. coli* showed highest MDR rates of 90% (9/10), 80% (8/10), 79.3% (23/29) and 79.2% (61/77) respectively (Table 6).

## Discussion

Antimicrobial resistance (AMR) is a critical global health challenge, threatening our ability to effectively treat infections and manage complications in health care facilities. In the current study, overall prevalence of culture-confirmed bacterial isolates was 41% (478/1165) from clinical specimens. This frequency is comparable to a study in Debre Markos (48.7%) [17]. However, the current study's rate was higher than findings from Southern Ethiopia (32.6%) [18], Addis Ababa (32.8%) [19], Gondar (21.6%, 14.8%) [20,21], Jimma (22.1%) [22], Ghana (37.8%) [23], Nigeria (29.6%) [24], and Yemen (43.2%) [24] respectively. However, this was lower than a study conducted in Bahir Dar (61.6%) [25], Gondar (83.9%) [26] and India (91.3%) [27]. The most possible explanation could be due to the difference in culture identification technique in the study population, the study design, geographical location, etiological agents, and infection prevention and control policies between regions and countries [2,13,24].

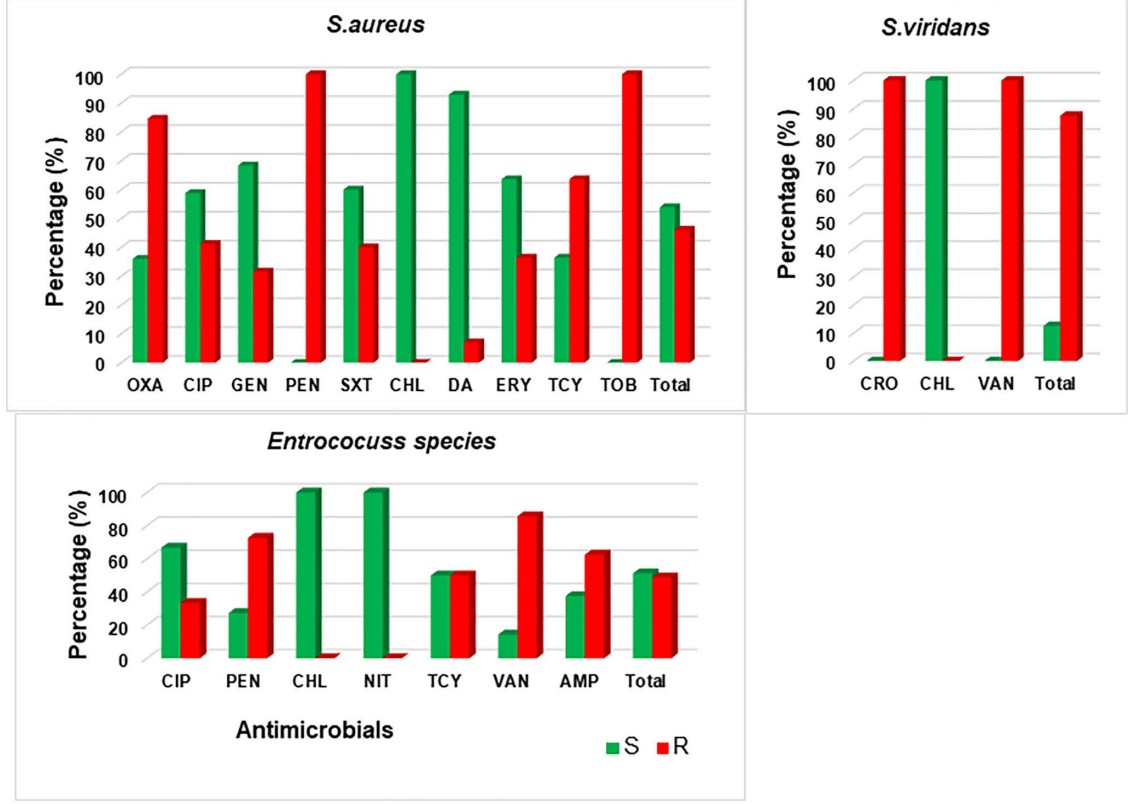

**Fig 2. Antimicrobial resistance levels of the three major top isolated Gram-positive bacteria.**

In this study higher proportion of Gram-negative bacteria 89.1% (426/478) compared to Gram-positive bacteria 10.8% (52/478) were isolated. Similarly, studies reported in Ethiopia (52.1% vs 47.9%) [28], (71.2% vs 28.8%) [29], (62.8% vs 37.2%) [30], (69% vs. 31%) [31], (57.6% vs 39.4% vs 3.0% [32], Tehran (72.2% vs. 27.8%) [33], Egypt (57.5% vs 31.1%) [34], India (68.1% vs 31.9%) [27] and Iran (55%) vs 45%) [35] revealed that the predominant isolates were Gram-negative bacteria. This might be due to differences in their cell wall structure, the presence of an outer membrane protein and the specific types of bacteria prevalent in the study population. However, another study reported from Ethiopia (87.7% vs 12.3%) [36] and (77.4% vs 22.6%) [37], India (53.0% vs. 39.0%) [38] and Tehran (64.2% vs 33.5%) [33] showed higher bacterial isolates caused by Gram-positive bacteria than Gram-negative bacteria respectively. The dominance of Gram-positive bacterial isolates over Gram-negative isolates in might be due to specific environment which the bacteria were isolated, the types of infections [20].

This study revealed that *Vibrio cholerae* 54.6% (261/478), *E. coli* 16.1% (77/478) and *S. aureus* 4.6% (22/478) were the most prevalent isolates. This is in line with a study conducted in Ethiopia, *S. aureus* was the predominant isolate (31.5%) followed by *E. coli* (13.8%) [17], Iran *E.coli* (7.58%), *Vibrio cholerae* (66%)) [39], India *E coli* (12.8%) and *S. aureus* (8.4%)) [27], Nepal (*S. aureus*, *E. coli* and *Vibrio cholerae*) were 68%, 53% and 6% [40], respectively. In addition, *E. coli* (9%) and *S. aureus* (44%) were isolated from Malaysia [41]. This might be due to conflict and civil unrest, refugees under poor conditions and fragile health infrastructure [42].

A high percentage of Gram-positive bacteria exhibit resistance to multiple antimicrobials, including tobramycin 100% (1), vancomycin 91.6% (11/12), oxacillin 87.5% (14/16), penicillin 83.3% (25/30), ceftriaxone 66.6% (2/3), and tetracycline 60%

**Table 5. Antimicrobial resistance profile of Gram-positive bacteria isolates.**

| Antimicrobial | *Enterococcus* spp | | *S. aureus* | | CONS | | *S. pyogenes* | | *S. viridians* | | Total | |
|---|---|---|---|---|---|---|---|---|---|---|---|---|
| | #T | R (%) | #T | R (%) | #T | R (%) | #T | R (%) | #T | R (%) | #T | R (%) |
| OXA | NA | NA | 13 | 11 (84.6) | 3 | 3 (100) | NA | NA | NA | NA | 16 | 14 (87.5) |
| CRO | NA | NA | 1 | 0 (0.0) | NA | NA | NT | NT | 2 | 2 (100.0) | 3 | 2 (66.6) |
| CIP | 3 | 1 (33.3) | 17 | 7 (41.2) | 3 | 2 (66.7) | NA | NA | NA | NA | 23 | 10 (43.5) |
| GEN | NT | NT | 19 | 6 (31.6) | 4 | 2 (50.0) | NA | NA | NA | NA | 23 | 8 (34.8) |
| PEN | 11 | 8 (72.7) | 17 | 17 (100) | 1 | 0 (0.0) | 1 | 0 (0.0) | NA | NA | 30 | 25 (83.3) |
| SXT | NA | NA | 15 | 6 (40) | 2 | 2 (100) | NA | NA | NA | NA | 17 | 8 (47.0) |
| CHL | 11 | 0 (0.0) | 7 | 0 (0.0) | 3 | 0 (0.0) | NT | NT | 1 | 0 (0.0) | 22 | 0 (0.0) |
| DA | NA | NA | 14 | 1 (7.1) | 2 | 0 (0.0) | NT | NT | NT | NT | 16 | 1 (6.3) |
| ERY | NA | NA | 11 | 4 (36.4) | NT | NT | 1 | 0 (0.0) | NT | NT | 12 | 4 (33.3) |
| NIT | 1 | 0 (0.0) | 4 | 0 (0.0) | NT | NT | NA | NA | NA | NA | 5 | 0 (0.0) |
| TCY | 2 | 1 (50.0) | 11 | 7 (63.6) | 1 | 1 (100) | 1 | 0 (0.0) | NA | NA | 15 | 9 (60.0) |
| TOB | NA | NA | 1 | 1 (100) | NT | NT | NA | NA | NA | NA | 1 | 1 (100.0) |
| AMP | 8 | 5 (62.5) | NA | NA | NA | NA | 1 | 0 (0.0) | NA | NA | 9 | 5 (55.5) |
| VAN | 7 | 6 (85.7) | NA | NA | NA | NA | NT | NT | 5 | 5(100.0) | 12 | 11 (91.6) |
| Total | 43 | 21 (48.8) | 130 | 60 (46.1) | 19 | 10 (52.6) | 4 | 0 (0.0) | 8 | 7 (87.5) | 204 | 98 (48) |

OXA- oxacillin, CRO- ceftriaxone, CIP-ciprofloxacin, GEN- gentamicin, PEN- penicillin, SXT- trimethoprim-sulfamethoxazole, CHL-chloramphenicol, DA- clindamycin, ERY-erythromycin, NIT-nitrofurantoin, TCY- tetracycline, TOB-tobramycin, AMP- ampicillin, VAN-vancomycin, NA-not analyzed (during antimicrobial susceptibility testing and reporting antimicrobial were selected based on CLSI, 2024 for each organism group but agents of proven test efficacy that show unacceptable in vitro test susceptibility and ineffective clinically should not be analyzed), NT- not tested (antimicrobial were not tested due unavailability of antimicrobial agents in bacteriology and mycology reference laboratory).

**Table 6. Multidrug resistance profiles of bacterial isolates.**

| Bacterial isolates | Degree of resistance | | | | | | | | | |
|---|---|---|---|---|---|---|---|---|---|---|
| | R0 n (%) | R1 n (%) | R2 n (%) | R3 n (%) | R4 n (%) | R5 n (%) | R6 n (%) | R7 n (%) | R 8 n (%) | MDR* n (%) |
| *Enterococcus* spp (n = 14) | 1 (7.1) | 5 (35.7) | 6 (42.9) | 2 (14.2) | 0 | 0 | 0 | 0 | 0 | 2 (14.2) |
| *S. aureus* (n = 22) | 3 (13.6) | 4 (18.1) | 3 (13.6) | 6 (27.3) | 3 (13.6) | 0 | 3 (13.6) | 0 | 0 | 12 (54.5) |
| CONS (n = 10) | 7 (70) | 0 | 1 (10) | 2 (20) | 0 | 0 | 0 | 0 | 0 | 2 (20) |
| *S. pyogenes* (n = 1) | 0 | 1 (100.0) | 0 (0.0) | 0 | 0 | 0 | 0 | 0 | 0 | 0 (0.0) |
| *S. viridians* (n = 5) | 0 | 3 (60.0) | 2 (40.0) | 0 | 0 | 0 | 0 | 0 | 0 | 0 (0.0) |
| *A. baumannii* (n = 10) | 0 | 2 (20.0) | 0 | 1 (10) | 3 (30) | 4 (40) | 0 | 0 | 0 | 8 (80) |
| *Citrobacter* spp (n = 6) | 2 (33.3) | 0 | 0 | 1 (16.6) | 2 (33.3) | 1 (16.6) | 0 | 0 | 0 | 4 (66.6) |
| *Klebsiella* spp(n = 29) | 1(3.4) | 2 (6.8) | 2 (6.8) | 5 (17.2) | 2 (6.8) | 5 (17.2) | 7 (24.1) | 3 (10.3) | 1 (3.4) | 23 (79.3) |
| *Enterobacter cloacae* (n = 3) | 0 | 1 (33.3) | 0 (0.0) | 0 | 0 | 1 (33.3) | 1 (33.3) | 0 | 0 | 2 (66.6) |
| *E. coli* (n = 77) | 2 (2.6) | 7 (9.1) | 7 (9.1) | 8 (10.4) | 13 (16.9) | 9 (69.2) | 11 (14.3) | 13 (16.9) | 7 (9.1) | 61 (79.2) |
| *P. aeruginosa* (n = 25) | 7 (28) | 5 (20) | 4 (16) | 6 (24) | 3 (12) | 0 | 0 | 0 | 0 | 9 (36) |
| *Proteus* spp (n = 10) | 1 (10) | 0 (0.0) | 0 (0.0) | 1(10) | 1(10) | 1(10) | 2 (20) | 1(10) | 3 (30) | 9 (90) |
| *Salmonella typhi* (n = 2) | 0 | 1 (50.0) | 1 (50.0) | 0 | 0 | 0 | 0 | 0 | 0 | 0 |
| *Vibrio cholerae* (n = 261) | 139(52.8) | 0 | 0 | 113 (43.3) | 9 (13.1) | 0 | 0 | 0 | 0 | 122 (46.7) |
| Others (n = 3) | 0 | 0 | 3 (100) | 0 | 0 | 0 | 0 | 0 | 0 | 0 |
| Total (n = 478) | 163 (34.1) | 31 (6.5) | 29 (6.1) | 145 (30.3) | 36 (7.5) | 21 (4.4) | 24 (5.1) | 17 (3.6) | 11 (2.3) | 254 (53.1) |

*MDR – Isolates resistant to 3 or more antimicrobial classes, R0- no antimicrobial resistant, R1 - resistant to one antimicrobial classes, R2 - resistant to two antimicrobial classes, R3- resistant to three antimicrobial classes, R4- resistant to four antimicrobial classes, R5-resistant to five antimicrobial classes, R6- resistant to six antimicrobial classes, R7 - resistant to seven antimicrobial classes, R8- resistant to eight antimicrobial classes.

(9/15). Similarly, high rates of resistant Gram-positive bacterial isolates reported in Debre Markos penicillin (89.7%) and tetracycline (71.3%) [17], Bahir Dar 65.4%, 42.6% and 34.6% were resistant to penicillin, tetracycline and oxacillin [43], Addis Ababa, penicillin (83.5%) and tetracycline (76.5%) [19], Egypt penicillin (89.5%) and oxacillin (76.52%) [20], Ruanda oxacillin (82.0%), penicillin (88%) and tetracycline (62%) [44], Malawi most bacteria exhibited high resistance to all commonly used antimicrobial excluding ciprofloxacin [45] and Nigeria penicillin (100%) [46] respectively. This concerning resistance is likely due to irrational use of antimicrobial and a lack of proper antimicrobial susceptibility testing in the region.

In the present study, Gram-negative isolates resistance to erythromycin 99.1% (116/117), ampicillin 94% (188/200), trimethoprim-sulfamethoxazole 82.1% (188/229), amoxicillin-clavulanic acid 81.5% (66/81) and ceftazidime 54.7% (80/146). This is in line with a study conducted in Debre Markos resistant to trimethoprim-sulfamethoxazole (53.1%) and ampicillin (70.4%) [17], Ceftazidime (77.2%) [36], Bahir Dar amoxicillin-clavulanic acid (90%) and ampicillin (85.7%) [43] and Gondar ceftriaxone (79.0%), trimethoprim-sulfamethoxazole(80.6%), amoxicillin-clavulanic acid (79.0%) were resistance [47].

Among Gram-negative bacterial isolates, *Klebsiella* spp resistance to ceftazidime 64% (16/25), ciprofloxacin 61.5% (5/8) and trimethoprim-sulfamethoxazole 76.2% (16/21). This is in line with a study conducted in Ethiopia resistant to ceftazidime (45%), ciprofloxacin (40%) and trimethoprim-sulfamethoxazole (45%) [48], trimethoprim-sulfamethoxazole (66.91%) [49], trimethoprim-sulfamethoxazole (100%) [50], trimethoprim-sulfamethoxazole (100%), ceftazidime (100%) and ciprofloxacin (90.9%) [51], ceftriaxone 43.3% [52], ceftriaxone, trimethoprim-sulfamethoxazole and tetracycline with a pooled resistance range of 40.6–55.3%) [52]. A study in Sudan showed that resistance to ceftazidime (95.4%) [53], Iraq ceftazidime with a resistance rate of 100% [54], Bangladesh (ciprofloxacin and trimethoprim-sulfamethoxazole) was 40% and 45% [48] and South Africa trimethoprim-sulfamethoxazole (50%) [55]. These findings indicate a serious challenge in treating infections caused by these bacteria, as many commonly used antimicrobials are not effective.

In this study, *Proteus* spp showed 85.5% (6/7) resistance to ceftazidime, 75% (6/8) to ciprofloxacin, and 75% (6/8) to trimethoprim-sulfamethoxazole. This is line with a study conducted in Debre Berhan ceftazidime (99%) [56], Gondar ceftazidime (46.7%) [3], Nekemte ceftazidime (100%) [57], Egypt trimethoprim/sulfamethoxazole (80.6%), amoxicillin-clavulanic (57.3%) and ceftazidime (55.3%) [58], Congo ciprofloxacin (78.6%) and trimethoprim-sulfamethoxazole (100%) [59] and Sierra Leone ciprofloxacin (50%) [60]. But on the contrary, studies done at Nekemte, none of the isolates were resistant to ciprofloxacin [57]. The World Health Organization (WHO) categorizes antibiotic-resistant Gram-negative bacteria, including *Proteus* spp, high-priority pathogens due to their significant threat to public health. These bacteria are often resistant to last-resort antibiotics like carbapenems and third-generation cephalosporins, leading to increased mortality rates [61].

In total, 53.1% (254/478) of the bacterial isolates were classified as multidrug-resistant (MDR). This finding is consistent with study findings reported from Ethiopia (64.2%) [62], (56%) [63], (88.8%) [64], 70% [18], (78.57%) [65], 78.2% [66], 77.9% [67], and (77%) [68]. A comparable result was reported in the studies conducted in India 50% and 66.1% [69,70], China (42.5%) [71], Tanzania (70.5%) [72], Sierra Leone (64.3%) [73], Ghana (89.5%) [74], and Egypt (65.5%) [75]. However, the current study finding was higher than the study conducted in India (37.1%) [76], Nepal (42.6%) [77], Australia (36%) [78], Indonesia (28.7%) [79], the USA (27%) [80], France (11.6%) [81] and Tanzania (43.0%) [82]. This difference might be due to many factors, including sample size, sites of infection, study area, infection prevention practices and improper use of antimicrobial [83,84].

## Limitations of the study

The limitation of this study was that bacterial species were identified by phenotypic methods. Due to the retrospective nature of the data, we did not investigate risk factors for bacterial infection and antimicrobial resistance profiles.

## Conclusions

This study highlights a significant prevalence of antimicrobial resistance among bacterial isolates in the specified area, with a notable 41.4% (478/1165) of specimens yielding positive cultures. Higher rates of resistance to the commonly used

antimicrobial agents were noticed for both Gram-negative and Gram-positive bacterial isolates. Moreover, MDR has been indicated in more than half of the bacterial isolates. Among bacterial isolates, a significant proportion 93.7% (238/254) of MDR were Gram-negative bacteria, which underscores the urgency of the situation. To effectively combat the issue of antimicrobial resistance, healthcare providers should prioritize judicious antibiotic prescribing practices, informed by local antibiogram data.

## Supporting information

**S1 File. Excel raw data.**
(XLSX)

## Acknowledgments

We would like to acknowledge the bacteriology and mycology reference laboratory expert at APHI and the regional public health research ethical review committee for giving me a chance to conduct research on this topic and for their approval.

## Author contributions

**Conceptualization:** Michael Getie, Alem Tsega, Tsehaynesh Gebreyesus, Gizeaddis Belay, Alemayehu Abate, Hailu Getachew, Bayeh Abera, Demeke Endalamaw, Tazeb Molla, Teshiwal Deress, Wudu Tafere, Belay Bezabih.

**Data curation:** Michael Getie, Alem Tsega, Tsehaynesh Gebreyesus, Gizeaddis Belay, Alemayehu Abate, Hailu Getachew, Bayeh Abera, Demeke Endalamaw, Tazeb Molla, Teshiwal Deress, Wudu Tafere.

**Formal analysis:** Michael Getie.

**Investigation:** Michael Getie.

**Methodology:** Michael Getie, Belay Bezabih.

**Resources:** Michael Getie.

**Software:** Michael Getie.

**Supervision:** Michael Getie, Bayeh Abera, Wudu Tafere, Belay Bezabih.

**Validation:** Michael Getie, Belay Bezabih.

**Visualization:** Michael Getie, Alem Tsega, Tsehaynesh Gebreyesus, Gizeaddis Belay, Alemayehu Abate, Hailu Getachew, Bayeh Abera, Demeke Endalamaw, Tazeb Molla, Teshiwal Deress, Wudu Tafere, Belay Bezabih.

**Writing – original draft:** Michael Getie.

**Writing – review & editing:** Michael Getie, Bayeh Abera, Belay Bezabih.

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
