## [Decision Letter · Decision Letter 0]

19 May 2025

Dear Dr. Getie,

Thank you for submitting your manuscript to PLOS ONE. After careful consideration, we feel that it has merit but does not fully meet PLOS ONE’s publication criteria as it currently stands. Therefore, we invite you to submit a revised version of the manuscript that addresses the points raised during the review process.

We look forward to receiving your revised manuscript.

Kind regards,

Tebelay Dilnessa, MSc

Academic Editor

PLOS ONE

Journal Requirements:

2. We require the following information in order to proceed with your submission: As you are reporting a retrospective study of medical records or archived samples, please ensure that you have discussed whether all data were fully anonymized before you accessed them and/or whether the IRB or ethics committee waived the requirement for informed consent. If patients provided informed written consent to have data from their medical records used in research, please include this information in the Methods section.

5. We are unable to open your Supporting Information file “Suporting information.rar”. Please kindly revise as necessary and re-upload.

Additional Editor Comments:

The paper generally requires intensive revision. That is, it requires a through edition, revision and proofreading in terms of typographically, punctuation and grammatically.

The background of the abstract was no explanatory.The abstract part of study period should be one and remove the other.Consistent use of the words/phrases such as drug, antimicrobial or antibiotics is necessary.It is better the author use standard form of Gram-positive or Gram-negativeIn the abstract and result, the absolute number (numerator and denominator) is needed together with the percentage. For example, A/B (C%).Write the name of the bacteria at its first appearance, no abbreviation is needed; then for the second appearance write the abbreviated form and continue. Here, please rewrite the abstract part.Revise also the conclusion based on the findingsThe introduction should be written in terms of burden of bacteria, rationality and objectives of the study.The discussion seems like result and requires revision.The discussion should be supported by reasons for variation or similarity, not merely comparison.Figures should be uploaded in separate file as ‘TIF’ versionFollow properly the manuscript writing protocol for PLoS one.

Reviewers' comments:

Reviewer's Responses to Questions

**Comments to the Author**

1. Is the manuscript technically sound, and do the data support the conclusions?

Reviewer #1: Yes

Reviewer #2: Yes

2. Has the statistical analysis been performed appropriately and rigorously?

Reviewer #1: Yes

Reviewer #2: Yes

3. Have the authors made all data underlying the findings in their manuscript fully available?

Reviewer #1: Yes

Reviewer #2: Yes

4. Is the manuscript presented in an intelligible fashion and written in standard English?

Reviewer #1: Yes

Reviewer #2: No

Reviewer #1: The author can see the detailed from the attached pdf

The authors should clearly explain what is known and unknown about the study?

On the methodology part you have you used cross sectional study design but I didn’t have seen the elaboration about sampling. Does your study is survey type?

On Operational definitions line 212 to 216 if you think the terms are already known by readers leave it no need of define. But if it is obligated the authors should cite the appropriate reference

Reviewer #2: Thank you for the opportunity to review the manuscript entitled "Antimicrobial resistance profiles of bacterial from clinical specimens in Amhara National Regional State Public Health Institute, Bahir Dar, Ethiopia: a retrospective study". This manuscript addresses the critical issue of antimicrobial resistance (AMR) in a high-burden setting, providing valuable surveillance data from a regional reference laboratory in Ethiopia. The topic is highly relevant globally, particularly for low- and middle-income countries (LMICs) where AMR surveillance is often limited. The retrospective study design and data from clinical specimens over a 2.5-year period offer insight into local resistance trends that can inform public health interventions and antimicrobial stewardship. However, while the study has merit, several aspects require major revision before it can be considered for publication.

Comments:

Comment 1 (General comment on the manuscript): The manuscript requires extensive language editing. Numerous grammatical issues and awkward phrasing obscure key points (e.g., “big concern” should be “serious concern”). The narrative lacks clarity in places, particularly in the results and discussion sections. Consider professional English editing.

Comment-2 (Title): Better if edited like as: Antibiotic Resistance Profiles of Bacteria Isolated from Clinical Specimens at Amhara National Regional State Public Health Institute, Bahir Dar, Ethiopia: A Retrospective Study. Antibiotic resistance" replaces "antimicrobial resistance" to specifically target bacterial pathogens and Bacteria isolated from clinical specimens" improves clarity and grammatical flow.

Abstract section:

Comment 3: The abstract is overly detailed for a structured abstract, (should be within 250-300 words).

Comment 4: Consider shortening results to key findings with precise percentages for major resistant pathogens. Ensure consistent formatting of species names (e.g., Escherichia coli, Staphylococcus aureus) in italics throughout. Do not use abbreviation and also both in the abstract.

Comment 5: In line number 36: “The drug susceptibility test was performed using the Kirby-Bauer disc diffusion method on Muller-Hinton agar” who was performed the AST? Please make it clear or omit. And also use the specific term like antibiotic rather drug.

Comment 6: Reduce redundancy of information, like study period, you have mentioned in both background and methods.

Comment 7: In the results section (lines 38–62), please include only the key findings with precise percentages for major resistant bacterial pathogens, and omit the fungal findings, as your study focuses on bacteria, not all microorganisms.

Comment 8: The conclusion effectively highlights the public health implications of the findings. However, consider strengthening the impact by specifying the most concerning resistant pathogens identified in your study. This would better tie the conclusion to your results and emphasize the urgency of surveillance and stewardship efforts.

Methods and Materials section: Please follow the PLOS ONE guideline:

Comment 9: Rewrite “Material and Methods” like as Methods and Materials

Comment 10: Break specifically “Study design, period and area” and maintain consistency

Comment 11: There is a lack of detail regarding quality control measures and breakpoint interpretation standards (e.g., CLSI 2021 or 2022?). Ensure that versions are cited and methodological adherence is clear.

Comment 12: Study population (line number132-136) “….a patients who visited APHI Medical Microbiology Reference Laboratory and had complaints of any infection suspected of microbial infections during the study period”. As you know your study was retrospective so make it clear to reduce confusion with prospective aspects.

Comment 13: Consider clarifying inclusion/exclusion criteria, e.g., how incomplete records were managed and whether duplicates were removed.

Comment 14: Line number 146 “Specimen collection, processing, and bacterial identifications” since your study was retrospective, thus, it is better to elaborate more about the recorded data, the way of collection and maintaining of the quality of data rather than stating about diagnostic process.

Comment 15: Please recheck and rewrite/rephrase methods and materials section by considering your source of data (line number 118-204)

Results section:

Comment 16: The results are comprehensive but not well-structured. Tables are very large and difficult to interpret.

Comment 17: Consider focusing on the most clinically relevant pathogens (e.g., E. coli, K. pneumoniae, S. aureus) in the main text and shifting exhaustive antibiograms to supplementary material.

Comment 18: Define key terms such as "MDR" clearly within the results, and consider reporting "XDR" and "PDR" if applicable.

Discussion section:

Comment 19: The discussion is lengthy, repetitive, and lacks a clear focus on clinical and policy implications.

Comment 20: There is limited critical reflection on factors contributing to resistance in the local context (e.g., prescribing practices, diagnostics access, stewardship programs).

Comment 21: Compare findings with WHO GLASS data and other sub-Saharan Africa studies more systematically.

Ethical clearance section:

Comment 22: Ethical approval is adequately described. However, the data availability statement must align with PLOS ONE policies, simply stating "data within manuscript" may not be sufficient.

**Do you want your identity to be public for this peer review?** For information about this choice, including consent withdrawal, please see our Privacy Policy

Reviewer #1: **Yes: ** Gebrie Kassaw Yirga

Reviewer #2: No

---

## [Author Response · Author response to Decision Letter 1]

3 Jul 2025

Yes, we address all comments raised by editors and reviewers

---

## [Decision Letter · Decision Letter 1]

25 Aug 2025

Dear Dr. Getie,

Thank you for submitting your manuscript to PLOS ONE. After careful consideration, we feel that it has merit but does not fully meet PLOS ONE’s publication criteria as it currently stands. Therefore, we invite you to submit a revised version of the manuscript that addresses the points raised during the review process.

We look forward to receiving your revised manuscript.

Kind regards,

Tebelay Dilnessa, MSc

Academic Editor

PLOS ONE

**Journal Requirements:**

**Additional Editor Comments:**

The paper was improved, still it requires revision. I would like to recommend you that you have to revise it properly.Line 104: Materials and methodsLines 60 &61: It is better the author use standard form of Gram-positive or Gram-negative (previous comment).Lines 106 and 107: The study was conducted at APHI, Bahir Dar city. Which is 565 km away from Addis Ababa (the capital of Ethiopia) a………Please revise it, based on the context, there is no sentence that starts by using ‘Which’.Write separately ‘study population’ and ‘sampling technique’.Line 144: Bacterial isolation and identificationLine 206: Data quality controlI recommend a separate heading as ‘Sociodemographic characteristics of study participants’ and ‘Magnitudes of bacteria’.Rewrite from line 251 to 267. A sentence cannot start just by saying ‘Table 1:……’Line 270: What does it mean ‘Table 2’?In table 2, you considered ‘***Candida albicans’***as bacteria. Is that correct? Please make a revision.In the abstract and result, the absolute number (numerator and denominator) is needed together with the percentage. For example, A/B (C%) (previous comment).Line 311 and 326: Gram-negativeLine 334, 346, etc: Gram-positive

Reviewers' comments:

Reviewer's Responses to Questions

**Comments to the Author**

Reviewer #3: (No Response)

2. Is the manuscript technically sound, and do the data support the conclusions?

Reviewer #3: Yes

3. Has the statistical analysis been performed appropriately and rigorously?

Reviewer #3: Yes

4. Have the authors made all data underlying the findings in their manuscript fully available?

Reviewer #3: Yes

5. Is the manuscript presented in an intelligible fashion and written in standard English?

Reviewer #3: Yes

**Reviewer #3: ** Article Review Report

Title: Antimicrobial Resistance Profiles of Bacteria from Clinical Specimens at Amhara National Regional State Public Health Institute, Bahir Dar, Ethiopia: A Retrospective Study

longer, reduce the length for more brevity

Abstract

Line 28: Replace "antimicrobial resistance pattern" with "antimicrobial resistance profile" for consistency with the title.

Line 32: Specify the age range of patients included in the laboratory results

Punctuation: Add a colon (:) after "Background" and "Methods."

Line 31: It is generally preferred to use "July 1, 2022, to December 31, 2024" instead of "1/7/2022 to 31/12/2024.

Line 36: If data were collected, please indicate the tools used; if the data were extracted rather than collected, please clarify that.

Line 37: In "descriptive statistics," it would improve clarity to specify the types of analyses conducted (e.g., frequency distributions, chi-square tests).

Line 40: add denominator for 41.4% (95% CI: 39; 44) and indicate in your mother document the section that show stool (263; 54.6%), urine (98; 20.3%), wound (62; 12.9%), blood (47; 9.8%).

In line 42-48: if you are reporting bacteria just use the total bacterial isolates which is 478

Methods and Materials

rewrite "Methods and Material" as: Methods and Materials

Study area

Include details about the type of culturing and the data handling and storing system to ensure consistency with WHONET,

In line 120-21, data records of two and a half years of data from 1/7/2022 to 31/12/2024; both times operationally similar. Use one of them.

In line 122-4, the objective is already stated under introduction, no need to repeat

In line 131-4, ….. The study utilized an entire sampling technique that included all available microbiology culture records of patients of all age groups suspected of having microbial infections during study period. What is that entire sampling technique?

Inclusion and exclusion criteria

In line 138, Patient seating indicates?

Bacterial identification

In line 151, indicate the year for CLSI

Data source and access

No comment

Specimen and Data quality control

No comment

Data analysis

Include my comment from abstract section

Ethics approval

In line 125-6: This manuscript does not report on or involve the use of any animal or human data or tissue. Be sure of this?

Results

Line 260: Please remove "Table 1" from the sentence "most frequently culture-processed specimens.

Culture Results in Table 2: Refine the percentages by calculating them based on the total sample denominator. Additionally, since the title focuses on antimicrobial resistance profiles of bacteria, consider excluding Candida albicans from the analysis or modifying the title accordingly.

Please provide a rationale for any variables that were not analyzed.

Discussion

Lines 381-382: The study reported an overall bacterial isolation frequency of 41.4% (482 isolates) from clinical specimens. However, this total includes 4 fungal isolates, which are not classified as bacterial isolates and need your action

In line 390: Better to move this explanation after line 390 ( The most possible explanation could be due to the difference in culture identification technique in the study population, the study design, geographical location, etiological agents, and infection prevention and control policies between regions and countries [2, 13, and 24]).

Reference at end of line 408 and 466

Reference

Check for correctness of reference for number 4(format), 7(format and version), 24 (format) and 46 (format)

**Do you want your identity to be public for this peer review?** For information about this choice, including consent withdrawal, please see our Privacy Policy

Reviewer #3: No

---

## [Author Response · Author response to Decision Letter 2]

23 Oct 2025

We are grateful for the opportunity to submit a revised draft of our manuscript entitled, " Antimicrobial Resistance Profiles of Bacteria from Clinical Specimens at Amhara Public Health Institute, Bahir Dar, Ethiopia: A Retrospective Study" to PLOS ONE. We also appreciate the time and effort you and each of the reviewers have dedicated to providing insightful feedback on ways to strengthen our paper. Thus, it is with great pleasure that we resubmit our article for further consideration. We have incorporated changes that reflect the detailed suggestions you have graciously provided. We also hope that our edits and the responses we provide below satisfactorily address all the issues and concerns you and the reviewers have noted.

To facilitate your review of our revisions, the following is a point-by-point response to the questions and comments delivered in your letter dated 24/8/2025.

---

## [Decision Letter · Decision Letter 2]

6 Nov 2025

Antimicrobial Resistance Profiles of Bacteria from Clinical Specimens at Amhara Public Health Institute, Bahir Dar, Ethiopia: A Retrospective Study

PONE-D-25-08930R2

Dear Dr. Getie,

We’re pleased to inform you that your manuscript has been judged scientifically suitable for publication and will be formally accepted for publication once it meets all outstanding technical requirements.

Kind regards,

Tebelay Dilnessa, MSc

Academic Editor

PLOS ONE

Additional Editor Comments (optional):

Reviewers' comments:

Reviewer's Responses to Questions

**Comments to the Author**

Reviewer #3: All comments have been addressed

2. Is the manuscript technically sound, and do the data support the conclusions?

Reviewer #3: Yes

3. Has the statistical analysis been performed appropriately and rigorously?

Reviewer #3: Yes

4. Have the authors made all data underlying the findings in their manuscript fully available?

Reviewer #3: Yes

5. Is the manuscript presented in an intelligible fashion and written in standard English?

Reviewer #3: Yes

Reviewer #3: Dear Authors,

I appreciate your thorough attention to the comments and the subsequent corrections made. Thank you for your dedication to this research, which significantly contributes to addressing the prevalent issue of antimicrobial resistance (AMR) in Ethiopia and beyond.

**Do you want your identity to be public for this peer review?** For information about this choice, including consent withdrawal, please see our Privacy Policy

Reviewer #3: No

---

## [Editor Report · Acceptance letter]

PONE-D-25-08930R2

PLOS ONE

Dear Dr. Getie,

I'm pleased to inform you that your manuscript has been deemed suitable for publication in PLOS ONE. Congratulations! Your manuscript is now being handed over to our production team.

Kind regards,

on behalf of

Dr. Tebelay Dilnessa

Academic Editor

PLOS ONE